# Multiple *ABCB1* transcriptional fusions in drug resistant high-grade serous ovarian and breast cancer

Elizabeth L. Christie[1,2], Swetansu Pattnaik[3], Jessica Beach[1], Anthony Copeland[1], Nineveh Rashoo[1], Sian Fereday[1], Joy Hendley[1], Kathryn Alsop[1], Samuel L. Brady [4], Greg Lamb[4], Ahwan Pandey[1], Anna deFazio[5,6,7], Heather Thorne[1], Andrea Bild [4,8] & David D.L. Bowtell[1,2,3]

*ABCB1* encodes Multidrug Resistance protein (MDR1), an ATP-binding cassette member involved in the cellular efflux of chemotherapeutic drugs. Here we report that ovarian and breast samples from chemotherapy treated patients are positive for multiple transcriptional fusions involving *ABCB1*, placing it under the control of a strong promoter while leaving its open reading frame intact. We identified 15 different transcriptional fusion partners involving *ABCB1*, as well as patients with multiple distinct fusion events. The partner gene selected depended on its structure, promoter strength, and chromosomal proximity to *ABCB1*. Fusion positivity was strongly associated with the number of lines of MDR1-substrate chemotherapy given. MDR1 inhibition in a fusion positive ovarian cancer cell line increased sensitivity to paclitaxel more than 50-fold. Convergent evolution of *ABCB1* fusion is therefore frequent in chemotherapy resistant recurrent ovarian cancer. As most currently approved PARP inhibitors (PARPi) are MDR1 substrates, prior chemotherapy may precondition resistance to PARPi.

[1] Peter MacCallum Cancer Centre, Melbourne 3000 VIC, Australia. [2] Sir Peter MacCallum Department of Oncology, The University of Melbourne, Parkville 3010 VIC, Australia. [3] Kinghorn Cancer Centre, Garvan Institute for Medical Research, Darlinghurst 2010 NSW, Australia. [4] The University of Utah, Salt Lake City, UT 84112, USA. [5] Centre for Cancer Research, The Westmead Institute for Medical Research, Westmead 2145 NSW, Australia. [6] Department of Gynaecological Oncology, Westmead Hospital, Westmead 2145 NSW, Australia. [7] The University of Sydney, Sydney 2052 NSW, Australia. [8] City of Hope, Los Angeles, CA 91010, USA. These authors contributed equally: Swetansu Pattnaik, Jessica Beach. Correspondence and requests for materials should be addressed to D.D.L.B. (email: d.bowtell@petermac.org)

Our previous whole-genome analysis of post-treatment high-grade serous ovarian cancer (HGSC) and breast cancer samples identified a transcriptional fusion between *ABCB1* and the upstream gene *SLC25A40* associated with up-regulation of *ABCB1* expression, while leaving the predicted MDR1 protein unaltered[1,2]. Here, we sought to determine the frequency and circumstances in which *ABCB1* fusions arise in a large cohort of recurrent HGSC and a smaller series of breast cancer patients.

## Results

**Prevalent over-expression of *ABCB1* in recurrent HGSC.** We analysed relapse ascites samples from 108 HGSC patients who had received a median of 2 (range: 0–9) lines of chemotherapy (Table 1 and Supplementary Data 1). We began by measuring *ABCB1* expression using quantitative polymerase chain reaction (Q-RT-PCR) and *SLC25A40–ABCB1*-specific PCR[1]. *SLC25A40–ABCB1* fusions were seen in 15.7% of recurrent HGSC samples (Fig. 1a). When rank ordered for *ABCB1* expression, fusions were distributed over 64 (59%) of the highest expressing samples. *ABCB1* over-expression has been observed in many tumour types[3], however, the threshold for calling clinically significant expression is unclear. The presence of *ABCB1* fusions across a wide span of tumours indicates that positive selection for MDR1 expression occurs in a majority of post-treatment HGSC samples.

**Table 1 Clinical characteristics of recurrent HGSC and breast cancer cohorts**

| Source | AOCS | | University of Utah | kConFab |
|---|---|---|---|---|
| *Age at diagnosis* | | *Gender* | | |
| Mean | 59.6 | Male | 0 | 1 |
| Range | 24.4–81.8 | Female | 13 | 19 |
| *Primary site* | | *Age at diagnosis* | | |
| Ovary | 77 | Mean | 54.6 | 45.2 |
| Peritoneum | 24 | Range | 34–83 | 33–61 |
| Female genital tract | 4 | *Grade* | | |
| Endometrium | 1 | 1 | 0 | 2 |
| Unknown | 2 | 2 | 3 | 6 |
| *Grade* | | 3 | 12 | |
| 1 | 3 | 4 | 1 | 0 |
| 2 | 12 | NA | 6 | 0 |
| 3 | 71 | *Phenotype at diagnosis* | | |
| Unknown | 22 | HR−, HER2− | 1 | 4 |
| *Stage* | | HR+, HER2− | 0 | 4 |
| I | 1 | HR+, HER2+ | 12 | 2 |
| II | 2 | HR−, HER2 NA | 0 | 4 |
| III | 80 | HR+, HER2 NA | 0 | 1 |
| IV | 13 | HR NA, HER2− | 0 | 1 |
| Unknown | 12 | HR NA, HER2+ | 0 | 1 |
| *Residual disease* | | NA | 0 | 3 |
| No macroscopic disease | 20 | *Progression free survival* | | |
| Macroscopic disease ≤1 cm | 36 | Number of events | 13[a] | 20 |
| Macroscopic disease >1 cm | 29 | Median months | 29 | 44 |
| Macroscopic disease, size unknown | 10 | 95% CI of median | 8–50 | 18–90 |
| Tumour not resected | 6 | NA | 1 | 0 |
| Unknown | 7 | *Overall survival* | | |
| *Progression free survival* | | Number of events | 13 | 18 |
| Number of events | 108 | Median months | 72 | 76 |
| Median months | 10.62 | 95% CI of median | 34–172 | 43–139 |
| 95% CI of median | 9.63–13.08 | *Germline mutation* | | |
| *Overall survival* | | BRCA1 | 0 | 4 |
| Number of events | 99 | BRCA2 | 0 | 3 |
| Median months | 31.3 | NA | 13 | 13 |
| 95% CI of median | 26.24–38.89 | *Sample collection procedure* | | |
| *Germline mutation* | | Biopsy | 0 | 1 |
| BRCA1 | 11 | Excision | 0 | 11 |
| BRCA2 | 2 | Autopsy | 0 | 8 |
| *Lines of prior chemotherapy* | | Thoracentesis | 13 | 0 |
| Median | 2 | *Prior treatment* | | |
| Range | 0–9 | Median lines chemotherapy | 5 | 1 |
| *Lines of prior MDR1 substrate chemotherapy* | | Range lines chemotherapy | 1–8 | 0–4 |
| Median | 2 | Median lines radiotherapy | 2 | 1 |
| Range | 0–6 | Range lines radiotherapy | 0–4 | 0–2 |
| | | Median lines targeted therapy | 0 | 0 |
| | | Range lines targeted therapy | 0–6 | 0–2 |
| | | Median lines hormone therapy | 2 | 0 |
| | | Range lines hormone therapy | 0–5 | 0–1 |
| *Total cases* | 108 | | 13 | 20 |

[a]One patient progressed but no date available
*NA* information not available

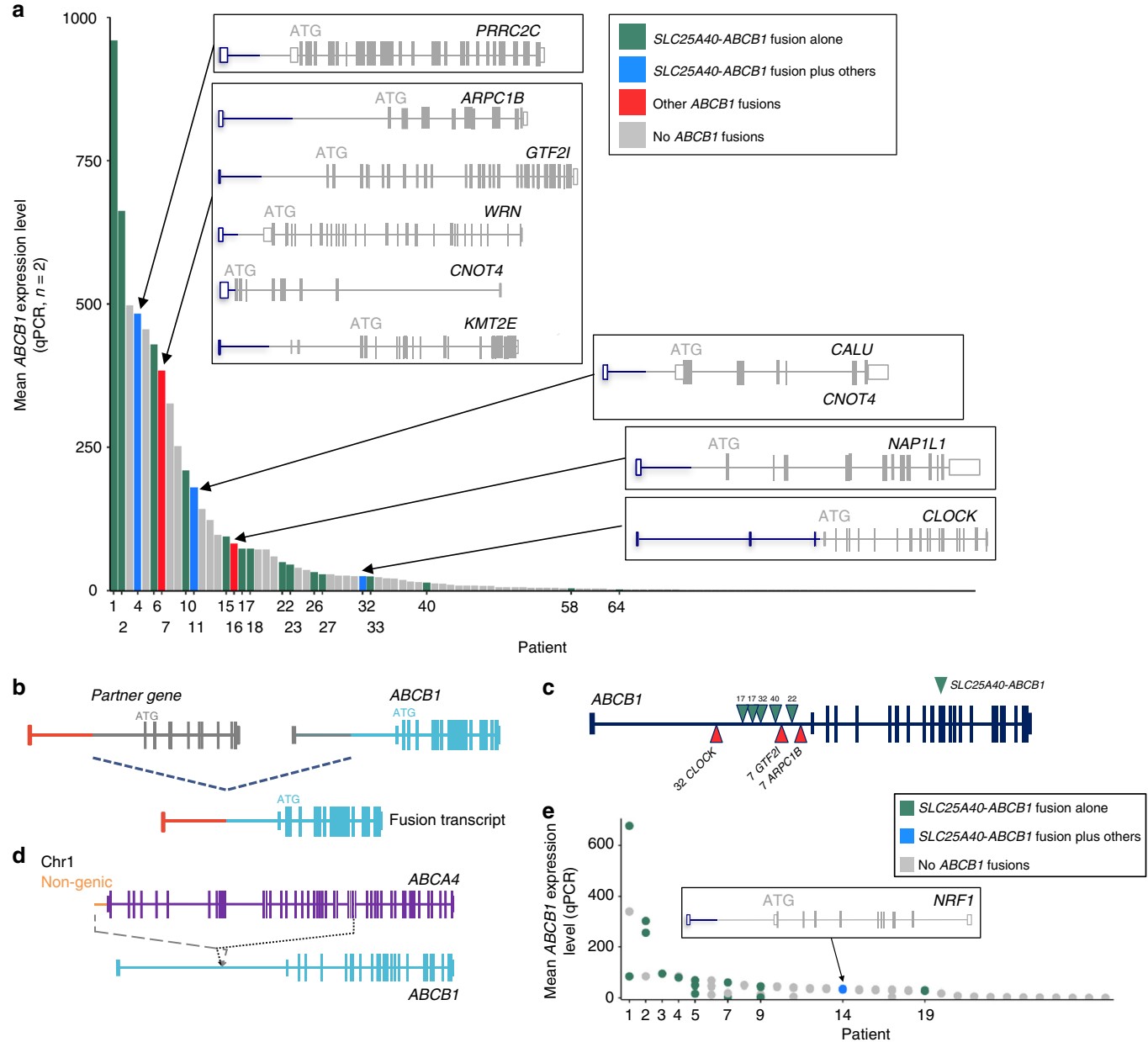

**Fig. 1** *ABCB1* in recurrent HGSC and breast cancer patients. **a** Patients ranked by *ABCB1* expression level in recurrent HGSC samples (*n* = 108), presence of *ABCB1* transcriptional fusions was observed in 20 patients, the bar colour indicates the fusions present. Gene structure of 10 of the 13 fusion partner genes is shown. **b** Schematic representation of the structure of a majority of the transcriptional fusions identified in which non-coding exons of partner genes (red) were fused to exon 2 onwards of *ABCB1* (green). **c** Clustering of breakpoints at the 3′ end of intron 1 of *ABCB1*, green triangles indicate *SLC25A40–ABCB1* fusion breakpoints, red are other fusion partners. **d** Schematic representation of the structure of the SV involving *ABCB1* in Patient 9. **e** Breast cancer patients ranked by *ABCB1* expression in recurrent or end-stage samples (*n* = 30 patients, 45 samples). *ABCB1* fusions were observed in nine patients (blue or green dots)

**Identification of multiple fusion partners with *ABCB1*.** In between samples with an *SLC25A40–ABCB1* fusion on the barchart were other tumours with high *ABCB1* expression without the *SLC25A40–ABCB1* fusion, suggesting that additional structural changes may deregulate *ABCB1*. We, therefore, characterised 25 recurrent HGSC samples with high levels of *ABCB1* using a modified 5′ Rapid Amplification of cDNA Ends (RACE) assay called FusionPlex[4]. We also performed whole-genome sequencing (WGS) and transcriptome analyses in ten samples that partially overlapped those subject to FusionPlex (Supplementary Data 1). We identified 16 novel *ABCB1* fusion partners (Supplementary Data 3). Twelve had the same general structure

as the *SLC25A40–ABCB1* event, involving breakage in intron 1 of *ABCB1* and splicing of non-coding 5′ exon(s) to exon 2 of *ABCB1* but leaving the predicted coding region unaltered (Fig. 1a, b, Table 2, Supplementary Data 3). Most of the novel fusions occurred in patients where a fusion with *SLC25A40* was also observed (Fig. 1a). Amongst the novel fusions only *CNOT4-ABCB1* was identified in more than one patient, suggesting that we had not screened to saturation and that other partners could be detected if further tumours were analysed.

Strikingly, Patient 7 had *ABCB1* fusions involving five different partners (Fig. 1a, Supplementary Data 3). In addition, as the breakpoints in *ABCB1* are unique to each translocation event we

**Table 2 Details of *ABCB1* transcriptional fusions**

| Cancer type | Case ID | Patient ID | Fusion | RT-PCR | WGS | RNAseq | FusionPlex |
|---|---|---|---|---|---|---|---|
| HGSC | 15245 | 1 | *SLC25A40:ABCB1* | Y | NT | NT | NT |
| HGSC | 65612 | 2 | *SLC25A40:ABCB1* | Y | NT | NT | NT |
| HGSC | 65554 | 4 | *MATR3:ABCB1* | NT | NT | NT | Y |
| HGSC | 65554 | 4 | *PRRC2C:ABCB1* | Y | NT | NT | Y |
| HGSC | 65554 | 4 | *SLC25A40:ABCB1* | Y | NT | NT | N |
| HGSC | 15224 | 6 | *SLC25A40:ABCB1* | Y | NT | NT | N |
| HGSC | 15340 | 7 | *ARPC1B:ABCB1* | Y | Y | Y | Y |
| HGSC | 15340 | 7 | *CNOT4:ABCB1* | NT | N | N | Y |
| HGSC | 15340 | 7 | *GTF2I:ABCB1* | N | Y | Y | Y |
| HGSC | 15340 | 7 | *KMT2E:ABCB1* | N | N | N | Y |
| HGSC | 15340 | 7 | *PHTF2:ABCB1* | NT | N | N | Y |
| HGSC | 15340 | 7 | *WRN:ABCB1* | N | N | N | Y |
| HGSC | 15272/AOCS-135 | 10 | *SLC25A40:ABCB1* | Y | N | N | N |
| HGSC | 15260 | 11 | *CALU:ABCB1* | Y | NT | NT | Y |
| HGSC | 15260 | 11 | *CNOT4:ABCB1* | Y | NT | NT | Y |
| HGSC | 15260 | 11 | *SLC25A40:ABCB1* | Y | NT | NT | Y |
| HGSC | 15335 | 15 | *SLC25A40:ABCB1* | Y | NT | NT | Y |
| HGSC | 15292 | 16 | *ITGB8:ABCB1* | NT | NT | NT | Y |
| HGSC | 15292 | 16 | *NAP1L1:ABCB1* | NT | NT | NT | Y |
| HGSC | 10335 | 17 | *SLC25A40:ABCB1* | Y | Y | Y | Y |
| HGSC | 10083/AOCS-117 | 18 | *SLC25A40:ABCB1* | Y | N | N | N |
| HGSC | 9954/AOCS-092 | 22 | *SLC25A40:ABCB1* | Y | Y | Y | N |
| HGSC | 12159 | 23 | *SLC25A40:ABCB1* | Y | NT | NT | N |
| HGSC | 65632 | 26 | *SLC25A40:ABCB1* | Y | NT | NT | NT |
| HGSC | 11424 | 27 | *SLC25A40:ABCB1* | Y | NT | NT | NT |
| HGSC | 10336 | 32 | *CLOCK:ABCB1* | Y | Y | Y | Y |
| HGSC | 10336 | 32 | *SLC25A40:ABCB1* | Y | Y | N | Y |
| HGSC | 15239 | 33 | *SLC25A40:ABCB1* | Y | NT | NT | NT |
| HGSC | 15209/AOCS_150 | 40 | *SLC25A40:ABCB1* | Y | Y | Y | N |
| Breast | | 14 | *NRF1:ABCB1* | NT | NT | NT | Y |
| Breast | | 19 | *SLC25A40:ABCB1* | Y | NT | NT | Y |
| Breast | | 21 | *TPX2:ABCB1* | NT | NT | NT | Y |

*Y* fusion identified, *N* fusion not identified, *NT* not tested

could determine that in Patient 17 two independent *SLC25A40* fusions to *ABCB1* had occurred (Fig. 1c, Supplementary Data 2). Overall, *ABCB1* fusions were identified in 18.5% of recurrent HGSC ascites samples tested. The presence of multiple different fusion partners, and multiple instances of the same fusion event in a given tumour, suggested a strong selective pressure for a convergent resistance phenotype in HGSC patients. It is notable that patient samples carrying ostensibly the same *SLC25A40–ABCB1* fusion event differed in the overall level of *ABCB1* expression, suggesting that fewer tumour cells carried the fusion event in ascites with lower *ABCB1* expression.

Structural variants (SV) involving *ABCB1* were detected in all ten samples subject to WGS (Supplementary Data 2), and verified three of the novel fusions identified by FusionPlex in two patients (Table 2). Patient 9's tumour sample very strongly expressed *ABCB1* but without a detectable fusion. We previously found a 95 kb insertion of part of the *ABCA4* gene in intron 1 of *ABCB1* of this tumour[1]. Analysis of RNAseq data showed strong expression from an alternative transcriptional start site in exon 2 of *ABCB1*, possibly due to the juxtaposition of regulatory elements from *ABCA4* (Fig. 1d). Therefore, SV in patient samples that did not result in predicted transcriptional fusion events may nevertheless alter *ABCB1* expression.

***ABCB1* fusions in end-stage breast cancer patients**. We extended our analysis to recurrent and rapid autopsy breast cancer patients ($n = 33$) (Table 1), a disease that shares common treatment approaches with HGSC. Using a combination of *SLC25A40–ABCB1* fusion-specific PCR and FusionPlex assays we detected *SLC25A40–ABCB1* fusions in nine patients. An *ABCB1*

fusion involving *NRF1* was found in Patient 14 that co-occurred with an *SLC25A40–ABCB1* fusion (Fig. 1e), demonstrating that convergent *ABCB1* deregulation also occurs in breast cancer patients. Patient 21 was observed to have a *TPX2–ABCB1* fusion by FusionPlex but did not have an *SLC25A40–ABCB1* fusion. The use of 18 autopsy samples from 6 fusion positive-breast cancer patients allowed us to obtain a survey of fusion positivity across metastatic deposits (Supplementary Fig. 18). Patient 5 was the only patient where all tested sites were *SLC25A40–ABCB1* fusion positive, Patients 1, 2, 7 and 9 had two fusion-positive sites and Patient 4 had only a single site that was positive for the *SLC25A40–ABCB1* fusion. The presence of subclonal fusion events suggests that in fusion-negative metastases novel mechanisms of *ABCB1* deregulation or other mechanisms conferring resistance are extant.

Amongst the HGSC and breast cancer samples, overall 15 fusions involving addition of non-coding exons 5′ from the partner gene to exon 2 of *ABCB1* were identified. Other ABC family transporters have been implicated in chemotherapy resistance[5], although the data is less compelling than for *ABCB1*. We evaluated samples where we had comprehensive WGS, copy number and transcriptome data but found little evidence of deregulation similar to that seen with *ABCB1* in other transporters (see Supplementary Information).

**Factors influencing selection of a fusion gene partner**. As *SLC25A40* was the most common fusion partner, and novel fusions with *ABCB1* were mostly found in tumours where an *SLC25A40–ABCB1* fusion was also detected, we considered why certain genes became partners to *ABCB1*. In principle,

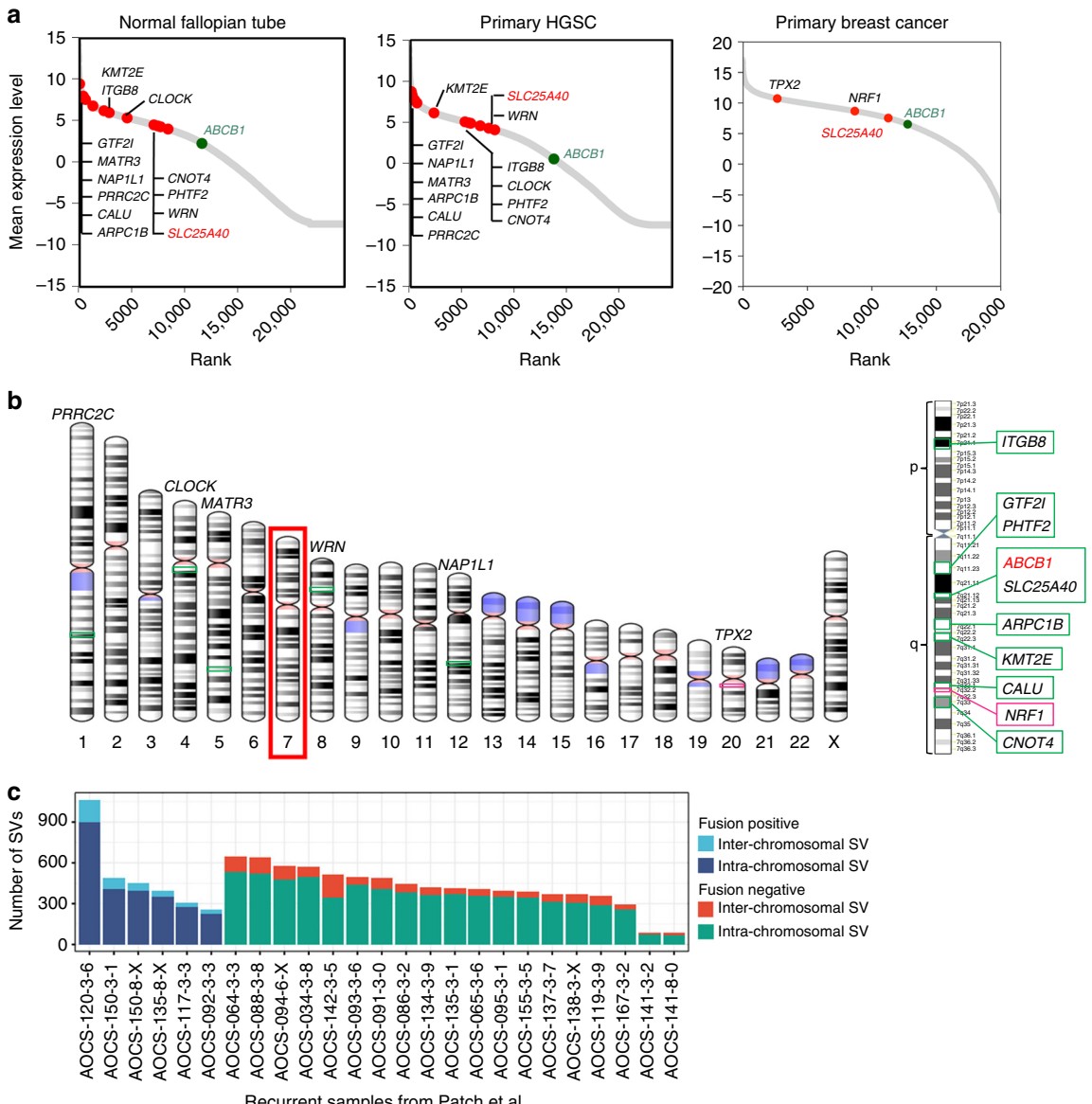

**Fig. 2** Characteristics of fusion partners. **a** Mean expression of all genes from normal fallopian tube, primary HGSC and primary breast cancer RNAseq data. Gene expression was rank ordered. Fusion partners are highlighted in red, *ABCB1* in green. **b** Chromosomal location of fusion partners, with a majority located on chromosome 7 (boxed). **c** Number of SVs, by type, in recurrent HGSC samples described as *ABCB1* fusion-positive or negative

transcriptional fusion could result in loss of negative regulation of *ABCB1* and/or acquisition of a constitutively expressed, stronger promoter in the lineage giving rise to HGSC. To assess promoter strength of the fusion partner genes we examined their RNA expression levels in pre-treatment tumour samples and fallopian tube secretory cells, the precursor cell type for HGSC. Consistent with fusion resulting in acquisition of a stronger promoter to drive *ABCB1* expression, the partner genes exceeded *ABCB1* expression in fallopian tube cells, and HGSC and breast cancer tumours obtained prior to chemotherapy exposure (Fig. 2a).

Surprisingly, *SLC25A40* was not the highest expressing gene amongst fusion partners, despite being the most consistently involved, suggesting that other factors must determine its selection. A majority of partner genes (9/15) are located on chromosome 7 (Fig. 2b). Amongst the fusion partner genes, *SLC25A40* was the physically closest downstream of *ABCB1*. While *RUNDC3B* is located closer to *ABCB1* than *SLC25A40* it is in the opposite transcriptional orientation to *ABCB1* and has

lower expression than *ABCB1* (Supplementary Fig. 19b). These findings suggest that chromosomal proximity to *ABCB1*, transcriptional level and orientation, and position relative to *ABCB1*, are major determinants in being involved in a productive fusion event, enabling cells to take advantage of simple deletions.

Our findings suggested that intrachromosomal rearrangements provided the highest probability of creating a productive *ABCB1* fusion. We considered whether tumours with a propensity to undergo structural rearrangement may be more likely to have fusion events, but we found no indication of an association between the frequency of SV in a sample and *ABCB1* fusion positivity (Fig. 2c). We, therefore, considered treatment-related factors that could influence the likelihood of emergence of *ABCB1* fusions. HGSC patients receive a wide spectrum of treatment regimens and their exposure to MDR1 substrate and non-substrate chemotherapies allowed us to relate the presence of a fusion to different treatment histories. Fusion events were only detected in patients who had been exposed to chemotherapies

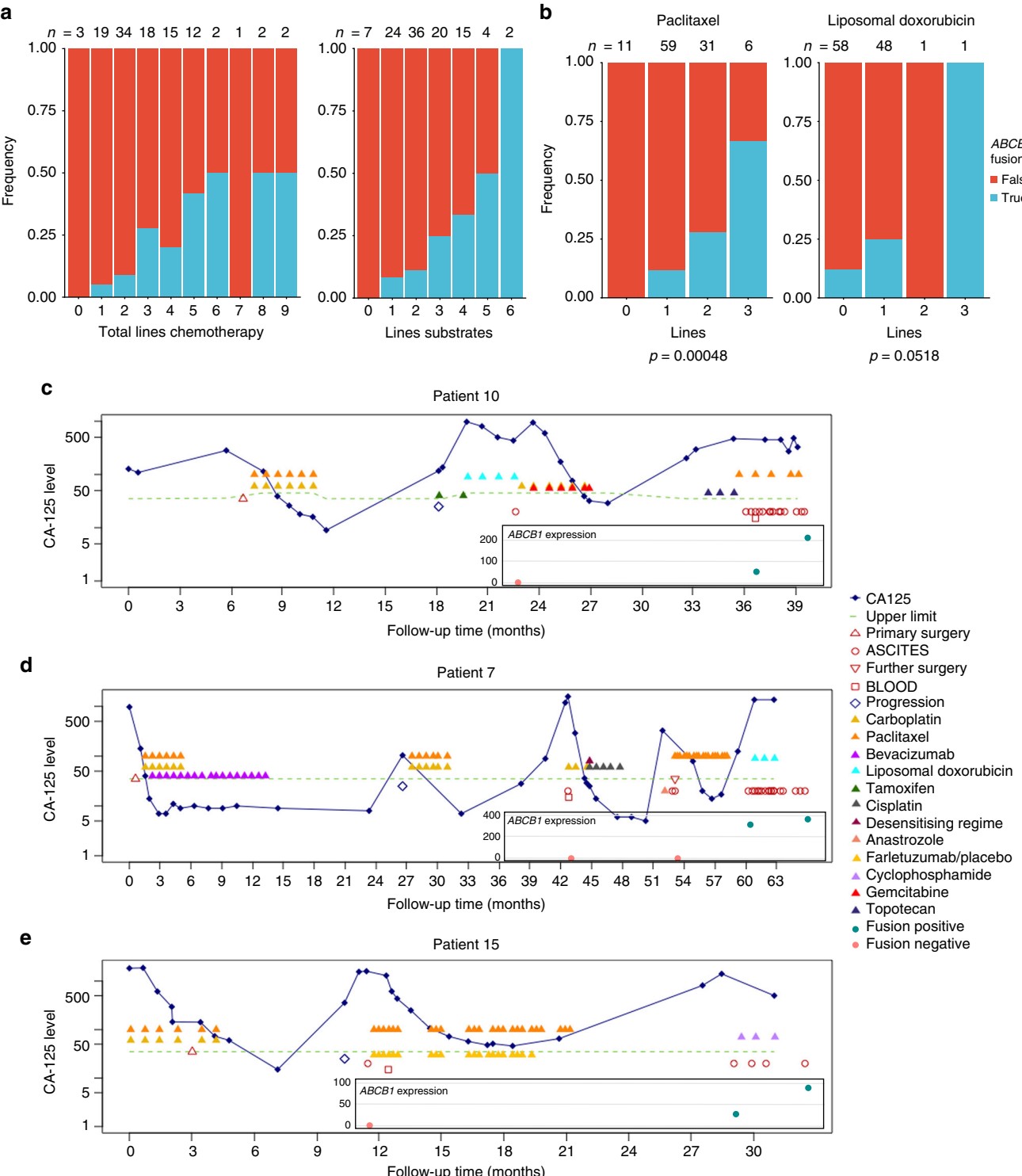

**Fig. 3** Association between treatment and fusion status. **a** Associations were identified between the total number of chemotherapy lines and number of MDR1 substrate chemotherapy lines and fusion positivity. **b** The number of lines of paclitaxel were significantly associated with fusion positivity, the number of lines of liposomal doxorubicin was not significant (Wilcoxon test). **c–e** CA125 serum marker profile for Patients 10, 15 and 7 where multiple recurrent ascites samples were tested for *ABCB1* expression (by Q-RT-PCR) and fusions, as indicated in graph inset

that are known substrates of MDR1, with the probability of a fusion event closely correlated to the number of lines of substrate chemotherapy (*p* value < 0.001, Wilcoxon test) (Fig. 3a). Of the 20 patients with fusion-positive tumours, 18 failed at least one line of treatment with a known MDR1 substrate (Supplementary Figure 20).

**Fusion positivity is associated with substrate chemotherapy**. We note that fusions were seen in two-thirds of patients who had received 3 lines of paclitaxel, and rose from 12 to 26% of patients following at least one line of liposomal doxorubicin (Fig. 3b): both drugs are MDR1 substrates and are known to induce chemoresistance by over-expression of MDR1[3]. However, given that

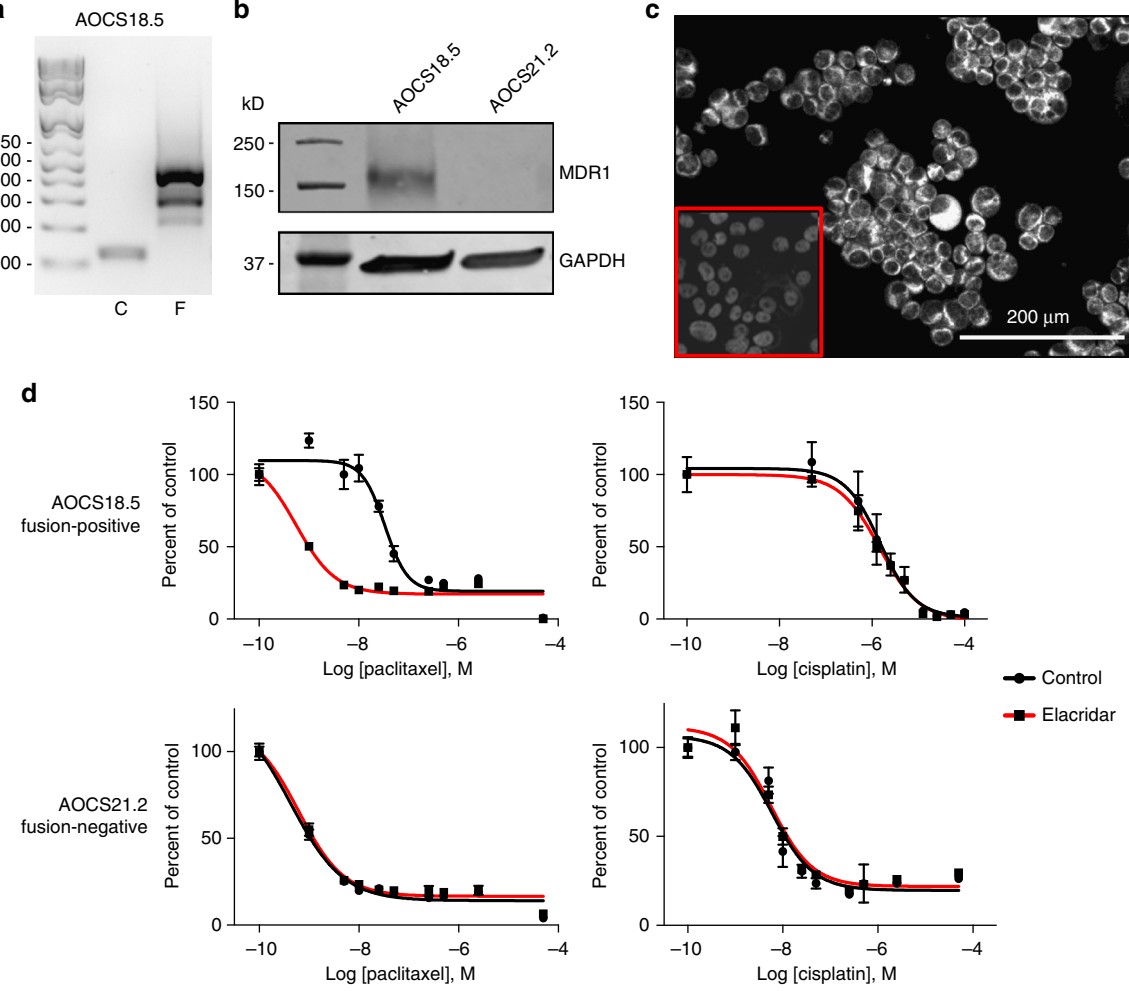

**Fig. 4** Resensitisation of a fusion-positive cell line with an MDR1 inhibitor. **a** Fusion-specific RT-PCR demonstrates that patient-derived cell line AOCS18.5 is positive for the *SLC25A40–ABCB1* fusion (F). C corresponds to RT-PCR using primers to measure *ABCB1* expression. Ladder is shown in base pairs. **b** Western blot shows MDR1 expression in fusion-positive line AOCS18.5 but not in fusion-negative line AOCS21.2. **c** MDR1 expression in AOCS18.5 by immunofluorescence. Inset shows antibody isotype control. **d** IC50 plots in AOCS18.5 and non-fusion AOCS21.2 cell lines, showing a fusion-specific increase in paclitaxel sensitivity, but sensitivity to non-substrate cisplatin, in the presence of the MDR1 inhibitor elacridar ($n = 3$ replicates). Error bars indicate ± SEM. For some points, error bars are shorter than the height of the symbol and are not shown

patients frequently received more than one line of a known MDR1-substrate chemotherapy prior to sample collection, it was difficult to determine which agent(s) were most likely to select for presence of a fusion. We, therefore, sought patients where samples were collected sequentially (Fig. 3c). We observed the appearance of a fusion-positive sample in Patient 10 following a line of carboplatin and gemcitabine followed by three cycles of topotecan. Carboplatin and gemcitabine are not effluxed by MDR1 but topotecan is a known MDR1 substrate[6], and therefore most likely to have resulted in the selection of the fusion in Patient 10. Consistent with the strong association of fusion positivity with number of lines of paclitaxel (Fig. 3b), we observed transition from negative to fusion positive in Patients 7 and 15 following paclitaxel treatment (Fig. 3c).

**Inhibition of MDR1 resensitizes to paclitaxel**. To further characterise the relationship between fusion positivity and paclitaxel sensitivity, and preclinically validate *ABCB1* fusions as a therapeutic target, we isolated a patient-derived HGSC cell line with the *SLC25A40–ABCB1* fusion (Fig. 4a). The line strongly expressed MDR1, as assessed by Western blot and immunocytochemistry (Fig. 4b, c). Elacridar is third generation MDR1

inhibitor that interferes with ATP hydrolysis by reducing ATPase activity[7]. Addition of elacridar potently re-sensitised the fusion-positive cell line to paclitaxel, but had no effect on its sensitivity to the non-substrate cisplatin (Fig. 4d), or on paclitaxel sensitivity of a fusion-negative HGSC cell line ($n = 3$ replicates) (Fig. 4d). Further functional characterisation of the impact of other promoter fusion partners on chemosensitivity awaits the availability of appropriate cell lines with these fusion events.

## Discussion

Our finding of frequent transcriptional fusions involving *ABCB1* in recurrent HGSC and breast cancer has important implications for chemotherapy choice in disease relapse and the clinical development of targeted agents. To our knowledge all currently clinically approved PARPi are MDR1 substrates and preclinically over-expression of MDR1 has been associated with olaparib resistance in cell line and animal models[8,9]. A requirement that patients should have received two to three lines of chemotherapy before being eligible for access to a PARPi may be unintentionally preconditioning tumours for resistance to these agents. Evaluation of fusion status may be particularly important in patients who have received dose-dense paclitaxel[10,11]. The clinical

development of novel targeted agents in HGSC should take into account whether the new agent is an MDR1 substrate and if so, determination made of fusion status prior to enrolment on trials of patients with recurrent disease. Future clinical trials could explore the benefit of stratifying patients based on fusion positivity and then directing fusion-positive patients to subsequent non-MDR1 substrate chemotherapy. Finally, although clinical findings with MDR1 inhibitors were largely disappointing, little or no patient stratification occurred[12], and it may be timely to reconsider previously abandoned MDR1 inhibitors[3].

## Methods

**Patients**. Ethics board approval was obtained at all institutions for patient recruitment, sample collection and research studies (Peter MacCallum Cancer Centre & University of Utah). Written informed consent was obtained from all participants in this study. Tables 1 and 2 summarises the clinical characteristics of each cohort.

The ovarian cancer cohort ($n = 108$) consisted of women diagnosed with epithelial ovarian, primary peritoneal or fallopian tube cancer since 1992. The women were treated at hospitals across Australia and were recruited through the Australian Ovarian Cancer Study (AOCS)[13]. All patients were diagnosed with serous carcinoma of high-grade (grade 2 or grade 3) and advanced stage (FIGO stage III or IV, International Federation of Gynaecology and Obstetrics) and received platinum-based chemotherapy as part of primary treatment. All samples utilised in the study were collected at relapse as ascites. Six of the 108 patients were previously described[1]. Summary treatment information is provided in Supplementary Data 1.

The kConFab cohort consisted of 20 breast cancer patients who were recruited to kConFab, a consortium for the study of familial breast cancer[14]. Patients have a strong family history of breast and/or ovarian cancer, 7 have germline BRCA1/2 mutations. Post-treatment samples were collected through biopsy ($n = 1$), excision ($n = 11$), or during rapid autopsy ($n = 8$).

Thirteen additional breast cancer patient samples were ascertained from Huntsman Cancer Institute, University of Utah. Samples were collected post-treatment during paracentesis.

**Nucleic acid isolation**. DNA was isolated from peripheral lymphocytes or lymphoblastoid cell lines using the salting out method for germline DNA. Tumour cells were isolated from ovarian cancer patient ascites using Dynabeads Epithelial Enrich (Invitrogen/Life Technologies), followed by DNA and RNA extractions. For breast cancer patients in the Bild cohort, cells from pleural effusions were collected through centrifugation and for 9 of 18 samples, CD45+, CD90+ and PDPN+ normal cells were depleted from the population using the Miltenyi quadroMACS as per manufacturer's protocol. Frozen tumour samples were cyrosectioned, $5 \times 50$ mm, for DNA and RNA extractions.

DNA extractions were performed using the DNeasy blood and tissue kit (QIAGEN), DNA was subsequently quantified using the Qubit dsDNA BR assay (ThermoFisher Scientific). For ovarian cancer and kConFab samples, RNA was isolated using the mirVana miRNA Isolation kit (Ambion/Life Technologies). The RNeasy Mini Kit (QIAGEN) was used to isolate RNA from Bild breast cancer pleural effusions. The Qubit RNA HS assay was used to assess RNA quantity.

**SNP arrays, tumour cellularity and copy number analysis**. Tumour and matched normal DNA was assayed with the Illumina HumanOmniExpress arrays as per manufacturer's instructions at the Australian Genome Research Facility (AGRF, Melbourne Australia). Single-nucleotide polymorphism (SNP) arrays were scanned on an iScan (Illumina), data was processed using the Genotyping module (v.1.9.4) in GenomeStudio v.2011.1 (Illumina) to calculate B-allele frequencies and logR ratios. Tumour cellularity was assessed using ASCAT[15] and qPURE[16], tumour ploidy was also calculated using ASCAT. Median tumour content was 96% (range: 70–99%).

**Whole-genome sequencing**. WGS libraries were generated from 1 µg of genomic DNA using TruSeq DNA PCR-free sample preparation protocol (Illumina, San Diego). Sequencing on a HiSeq X Ten (Illumina at GenomeOne, Kinghorn Comprehensive Cancer Centre, Sydney Australia) was performed to a minimum average of 30-fold base coverage for germline samples and 60-fold coverage for ascites samples (Supplementary Data 2).

Each lane of sequencing data underwent alignment to the Genome Reference Consortium human genome assembly (GRCh37) using BWA-MEM[17]. Optical duplicate reads were marked using Picard MarkDuplicates ([https://broadinstitute.github.io/picard/]).

The Genome Modelling System (GMS)[18] was utilised to perform end to end analysis of WGS data to call somatic SVs. The GMS workflow was configured to utilise a combination of two complementary algorithms—Breakdancer[19] and Squaredancer ([https://github.com/ding-lab/squaredancer]) followed by a read assembly step using TIGRA ([http://bioinformatics.mdanderson.org/main/

TIGRA]). Concurrently, Genome Rearrangement IDentification Software Suite (GRIDSS)[20] was utilised for its ability to detect SV at low cellular prevalence. GRIDSS was also run for the six samples previously described[1]. All SVs in ABCB1 were manually reviewed in integrative genomics viewer (IGV)[21].

**Transcriptome sequencing**. One microgram of total RNA was used for library preparation using the TruSeq RNA Sample Preparation kit (Illumina) as per the manufacturer's low-throughput protocol. All libraries were sequenced as paired-end 100 bp on a HiSeq2500 (Illumina, at AGRF), generating 100 million paired reads per sample.

Each barcode separated lane of sequencing data was aligned to the Genome Reference Consortium human genome assembly (GRCh37) using HISAT2[22]. RNA-seQC[23] was used to investigate RNA sequencing quality. Estimation of gene abundance was carried out in R using Rsubread[24], edgeR[25] and Limma[26]. Fusion transcripts from RNAseq data were detected using JAFFA[27] and STAR-Fusion[28]. A minimal criteria of ≥1 one split read support was imposed for filtering of predicted ABCB1 fusion events to facilitate the discovery of fusion transcripts with low representation in the transcriptome. All fusions involving ABCB1 were manually reviewed in IGV.

**Archer FusionPlex**. A custom Archer FusionPlex assay (ArcherDX, Colorado) was utilised to interrogate 5′ fusion partners for ABCB1. Up to 200 ng of RNA was utilised to generate sequencing libraries, as per manufacturer's protocol. Libraries were sequenced on a MiSeq (Illumina) generating paired-end 150 bp reads, RNA-seQC was used to examine sequencing and alignment quality. On average 2.1 million reads were generated per sample (range: 21,888–7,568,196). As per transcriptome sequencing, data was aligned to GRCh37 and fusions were detected with JAFFA. All reads that mapped to ABCB1 exons 2 and 3 and their pairs were manually reviewed in IGV.

***ABCB1* RT-PCR**. RNA was reverse transcribed into cDNA using random primers (Promega) and M-MLV reverse transcriptase (Promega).

Quantitative reverse transcription PCR (Q-RT-PCR) was used to measure ABCB1 transcript abundance. Q-RT-PCR was performed in triplicate to examine ABCB1 expression, with GAPDH or HPRT and ACTB for normalisation. Primer sequences are listed in Supplementary Table 1. The ΔΔCt method was used to calculate expression levels compared to ABCB1 expression in the SKOV3 cell line.

Testing for the presence of the SLC25A40–ABCB1 fusion transcript was performed using nested RT-PCR with primers to exon 1 of SLC25A40 and exon 3 of ABCB1, an assay described previously[1]. Primer sequences are listed in Supplementary Table 1. Briefly, PCR conditions were as follows: 98 °C for 30 s, 30 cycles of 98 °C for 10 s, 60 °C for 30 s, 72 °C for 10 s and 72 °C for 10 min. PCR product from first PCR was purified using QIAquick PCR purification kit (QIAGEN), for use as template in the second PCR. The PCR product from the second PCR was run on a 2% agarose gel.

Validation of novel fusion transcripts was performed using primers specific to each fusion partner and exon 3 of ABCB1 using the PCR conditions above (Supplementary Table 1).

**In vitro studies**. Patient-derived cell lines AOCS18.5 and AOCS21.2 were established from AOCS patient ascites. Approximately, 2 mL of ascites fluid collected at recurrence was centrifuged at 1500 rpm for 5 min to create a cell pellet. Supernatant was removed and the cell pellet was resuspended in 10 mL of complete RPMI media (RPMI 1640, 10% FBS, 50 µ/mL penicillin and 50 mg/mL streptomycin) and transferred into a standard humidified incubator (37 °C, 5% CO$_2$). Media were replaced after 48 h, and then once every 2–4 days, until an adherent cell line was established. Cells were passaged ten times from the time of collection and stocks cryopreserved. Both the AOCS18.5 and AOCS21.2 cell lines were authenticated against the patient germline DNA using STR profiling (GenePrint 10 System, Promega) and shown to be free of Mycoplasma via PCR prior to being used for in vitro studies (testing date: 27 July 2017)[29]. RT-PCR for the SLC25A40–ABCB1 fusion was performed on 20 cell lines, and one fusion-positive line was identified: AOCS18.5 (Fig. 4a). A fusion-negative line, AOCS21.2, was used as a control. Cell lines are available upon request.

Whole-cell lysates were prepared using RIPA lysis buffer containing protease inhibitor cocktail (Roche) followed by sonication. Protein concentrations were determined using the DC$^{TM}$ protein assay (Bio-Rad). Totally, 40 µg of lysate were loaded onto a 4–20% gradient polyacrylamide gel (Bio-Rad) and subjected to gel electrophoresis at 150 V for 1 h and membrane transfer was performed using the Trans-Blot Turbo Transfer System (Bio-Rad) using the High Molecular Weight protocol[30]. Membranes were blocked in Odyssey Blocking Buffer TBS (Li-Cor) for 1 h and then incubated with the appropriate antibody overnight at 4 °C. Antibodies included: MDR1 (1:1000, Abcam, ab170904) and GAPDH (1:10,000, Abcam, ab8245). Following overnight incubation, the membranes were washed three times with TBS-T and then incubated with the appropriate IRDye® secondary antibody (1:15,000, Li-Cor, 926–32,211 or 926–68,070) for 1 h at room temperature (RT). Membranes were washed three times with TBS-T and the signal was analysed by the Li-Cor Odyssey system.

Cytospins of AOCS18.5 cells were generated and fixed in 4% PFA for 1 h at RT. Cells were permeabilised using 0.1% Triton-X in 50 mM Tris-HCl pH7.6 at RT for 10 min, before washing in phosphate-buffered saline (PBS). Peroxidase activity was quenched with 3% $H_2O_2$ (5 min at RT), followed by PBS rinses. Slides were blocked for 30 min at RT in TNB Blocking Buffer (Anti-Rabbit Immpress Kit, Vector Labs). Anti-MDR1 antibody (1:400) (D3H1Q, Cell Signalling Technology) and Rabbit IgG Isotype control (1:145.5) in TNB Blocking Buffer were incubated on slides for 1 h at RT in a humid chamber. Slides were washed in TBST and incubated with PE Rabbit HRP (1:1000) for 30 min, before washing in TBST. Slides were incubated for 6 min with TSA reagent with 1:100 dilution of Tyramide fluor in PE Amplification Diluent, then washed in TBST. DAPI was added prior to coverslipping and imaging on the Vectra.

AOCS18.5 and AOCS21.2 cells were seeded in triplicate at $7.5 \times 10^3$ cells per well in 96-well black-walled plates 24 h prior to drug treatment. Cells were treated for 72 h with a 10-point dilution series of paclitaxel or cisplatin with or without 250 nM elacridar (Selleckchem) in antibiotic-free media. Cell viability was determined via DAPI staining and high content imaging on the Cellomics ArrayScan Vti platform. Briefly, cells were fixed with 4% PFA then permeabilized with 0.2% Triton X and stained with DAPI (1:1000 dilution). IC50 doses of paclitaxel and cisplatin were approximated by fitting a four-parameter dose–response curve (Hill equation) and all parameters used in curve comparison using Prism 7 (GraphPad)[31].

**Reporting summary**. Further information on experimental design is available in the Nature Research Reporting Summary linked to this article.

## Data availability

The whole-genome and transcriptome sequencing data will be deposited in the European Genome-phenome Archive (EGA). Most of the data will be publicly available, the germline data will not be publicly available due to restraints imposed by the ethics committee, requests for further data can be made to the EGA Data Access Committee (DAC).

The Patch et al. data is available from the EGA repository under the accession code EGAD00001000877. Primary triple negative breast cancer expression data ($n = 123$) was obtained from cBioPortal ([http://www.cbioportal.org/], Breast Invasive Carcinoma (TCGA, Nature 2012)).

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

## Acknowledgements

This study was financially supported by grants from the Department of Health and Human Services through the National Health and Medical Research Council of Australia (NHMRC, APP631701 and APP1124309), the US National Cancer Institute U54 programme (U54CA209978), Victorian Cancer Agency (ECSG15012) and Cancer Australia (CA, APP1004673). We gratefully acknowledge additional support from Mrs. Margaret Rose AM and the Rose family, The WeirAnderson Foundation, Border Ovarian Cancer Awareness Group, donors to the Garvan Institute of Medical Research's Ovarian Cancer Research Programme, Wendy Taylor and Arthur Coombs and family. The Australian Ovarian Cancer Study (AOCS) was supported by the U.S. Army Medical Research and Materiel Command under DAMD17-01-1-0729, The Cancer Council Victoria, Queensland Cancer Fund, The Cancer Council New South Wales, The Cancer Council South Australia, The Cancer Foundation of Western Australia, The Cancer Council Tasmania and the NHMRC (ID400413 and ID400281). We acknowledge the vital role of the Australian Ovarian Cancer Study and kConFab for this study. AOCS acknowledges additional support from Ovarian Cancer Australia and the Peter MacCallum Cancer Centre Foundation, and acknowledges the cooperation and contribution of the participating institutions in Australia and study nurses, research assistants and all clinical and scientific colla-borators including Nadia Traficante, Linh Nguyen, Gillian Mitchell, Margot Osinski, Karen Sanday, Helen Steane and Leanne Bowes. The complete AOCS Study Group can be found at [www.aocstudy.org]. CASCADE acknowledges support from the Peter MacCallum Cancer Centre Foundation. We wish to thank Eveline Niedermayr, all the kConFab research nurses and staff, the heads and staff of the Family Cancer Clinics, and the Clinical Follow Up Study, which has received funding from the NHMRC, the National Breast Cancer Foundation (NBCF), CA, and the National Institute of Health (USA) for their contributions to this resource, and the many families who contribute to kConFab. kConFab is supported by a grant from the NBCF, and previously by the NHMRC, the Queensland Cancer Fund, the Cancer Councils of New South Wales, Victoria, Tasmania and South Australia, and the Cancer Foundation of Western Australia. A. deFazio is supported by the University of Sydney Cancer Research Fund and Cancer Institute NSW, through the Sydney West Translational Cancer Research Centre. J. Beach was supported by the American

Australian Association Sir Keith Murdoch Fellowship. We would like to thank all of the women who participated in the study.

## Author contributions

Project supervision: A.B. and D.D.L.B. Study design: E.L.C. and D.D.L.B. Sample acquisition: S.F., K.A., J.H., S.B., G.L., Ad.F., H.T., A.B. and D.D.L.B. Sample preparation: E.L.C., J.B., A.C., N.R., J.H. and S.B. Sequence data management, alignment and mutation identification: E.L.C., S.P. and A.P. Data acquisition: E.L.C., J.B., A.C., N.R., S.F. and G.L. Data analysis: E.L.C., S.P., J.B. and A.P. Wrote the paper: E.L.C., A.B. and D.D.L.B.

## Additional information

**Competing interests:** D.B. receives research grant funding from Genentech, Roche & Astra Zeneca, and is establishing a research collaboration with BeiGene. BeiGene has developed pamiparb which is a non-P-gp substrate PARPi. The remaining authors declare no competing interests.

