## [Peer Review File · Nature Communications]

Reviewers' comments:

Reviewer #3 (Remarks to the Author):

I reviewed the original submission to Nature Genetics, and that review is included here for completeness.

The authors have addressed most of the points raised in the original review.

Original major points

Point 1. Relationship to key genomic features in HGSC including BRCA1/2 mutation and CCNE1 amplification.

Table 1 presents the number of patients with mutations in BRCA1 and BRCA2 (11 and 2 respectively). However, it is not clear if BRCA1/2 mutation status was known for all 108 patients or whether some were unknown? SI Table 6 suggests that only 6 BRCA1/2 wild-type cases were analysed by WGS? However, important to know if germline WT status was known for a larger number.

In addition, the results suggest no obvious correlation between presence of structural variants and presence of ABCB1 fusions – what about other key features of the HGSC genome, such as tandem duplicator phenotype and/or presence of fold-back inversions?

Point 2. Allelic frequency and clonality.

The increase in ABCB1 transcription between diagnosis and relapse generally suggests that there is selection for ABCB1-expression cells following chemotherapy. However, this does not indicate that ABCB1 in general and the fusions specifically are the drivers of relapse/resistance. I realise that the fusions were mainly identified via qRT-PCR and RACE, but is it possible calculate the allelic frequency of these fusions? Perhaps the rapid autopsy samples from breast cancer patients would make this simpler.

Points 3 and 4. Cell line models.

The authors have not re-expressed fusion cDNA in MDR1-null cells to confirm the functional effect. However, the cell lines depicted in 4D suggest that the fusions can be functional. However, it is not clear which cases these cells lines derive from (and thus precisely which fusion is expressed in AOCs18.5 cells). This is not clear from the nomenclature nor supplementary information. However, these cells should act as an important model for acquired drug resistance.

Original minor points

Both of these have been addressed.

Original review

In this study, the authors investigate the presence of ABCB1 fusions in patients with relapsed high grade serous ovarian cancer (N=108) or recurrent breast cancer (N=33). This publication arises on the back of the original description of SLC25A40-ABCB1 fusions in these authors ICGC Nature paper in 2015 – in that publication, 6 AOCs and three other patients were described.

The material assessed in the ovarian cancer cohort was ascites, and the data suggest that ABCB1 fusions are common (18/108) in recurrent ovarian cancer, with the majority involving SLC25A40. Several patients had multiple separate fusions, and the majority of fusions involved other genes on chromosome 7. There was a statistically significant correlation between number of lines of prior therapy and presence of fusions and, in particular, the number of lines of MDR1-substrate chemotherapy and presence of fusions, suggesting that these chemotherapy agents were strong inducers of fusions.

There were also ABCB1 fusions identified in breast cancer, at a frequency (9/33 patients) that may be higher than in HGSC (although the difference is not significant by my calculation).

Overall, the data are interesting but largely confirmatory of the original description in 2015 – this publication fills in some details, such as firming up the likely frequency, the association with number of lines of prior therapy and the location of other fusion genes within the genome but does not present major new findings. However, I do agree with the authors that MDR1 inhibitors should be re-assessed in light of these findings.

Several points arise:

1. Although there was no obvious correlation with number of SV overall in the HGSC genomes, was there any association with other known genomic events such as BRCA1/2 mutation, CCNE1 amplification etc?
2. All the ovarian cancer analyses were performed on cells isolated from ascites, which are likely to represent a mixture from multiple deposits within the peritoneal cavity – did the authors

have access to any multi-site solid tumour deposits to assess the intra-patient heterogeneity of fusion?

3. Have the authors attempted to re-express the fusion cDNA within MDR1-null cells to identify those most likely to induce drug resistance?
4. Does any of the fusions exist in established HGSC cell lines and thus act as a model of acquired drug resistance?

Minor points

1. Table 1 – need number of prior lines of therapy
2. Second paragraph of main text – I think that the number of prior lines should be presented as median rather than mean.

Reviewer 3:

The authors have addressed most of the points raised in the original review. It is not clear if BRCA1/2 mutation status was known for all 108 patients or whether some were unknown? SI Table 6 suggests that only 6 BRCA1/2 wild-type cases were analysed by WGS? However, important to know if germline WT status was known for a larger number. We have clarified the proportion of patients for whom BRCA1/2 germline status was known in Supplementary Information Section 6. In addition, the results suggest no obvious correlation between presence of structural variants and presence of ABCB1 fusions - what about other key features of the HGSC genome, such as tandem duplicator phenotype and/or presence of fold-back inversions?

Figure 2C shows the relationship between structural variants (SV) and presence/absence of a fusion. Since all tumours have some SVs, the association could be investigated in the relatively small number of patients where comprehensive genomic information was available. However, as only a subset of tumours have foldback inversions or TDs, it isn't possible to do a meaningful analysis of these additional structural aberrations and fusion positivity.

Point 2. Allelic frequency and clonality. The increase in ABCB1 transcription between diagnosis and relapse generally suggests that there is selection for ABCB1-expression cells following chemotherapy. However, this does not indicate that ABCB1 in general and the fusions specifically are the drivers of relapse/resistance.

The Reviewer's suggestion that we base our assertion of an association of ABCB1 with acquired resistance on an increase in expression is incorrect and overlooks many elements of the paper. Rather, we base our conclusions on multiple lines of orthogonal data: the appearance of different fusion partners with the same structure only at recurrence, including multiple fusion events in the same patient, and their very strong association with the number of lines of substrate chemotherapy (eg. $p > 0.001$ for paclitaxel). Much of the recent high-level literature from investigators such as Peter Campbell, Gaddy Getz, Serena Nik-Zainal, or Charlie Swanton involving the identification of driver mutations in various cancers have rested on a statistical enrichment of specific mutations, mostly without any functional data. So, statistical measures are a legitimate indicator of causality. However, we do also provide functional data in addition to our statistical argument, showing the effect of attenuation of the fusion on cell line sensitivity specifically to substrate chemotherapy and specifically in cells with the fusion (Figure 4).

Overall, we contend that our evidence for fusions being drivers of resistance equals or surpasses, for example, the evidence supporting the role of BRCA reversion in acquired resistance, which is not contested.

I realise that the fusions were mainly identified via qRT-PCR and RACE, but is it possible calculate the allelic frequency of these fusions? Perhaps the rapid autopsy samples from breast cancer patients would make this simpler.

As the Reviewer indicates, one can't use RNA data to calculate allelic frequency as the fusion increases expression of ABCB1. As shown in Table 2, the breast patients did not undergo WGS. As described in the text, the autopsy samples show only a proportion of the tumour sites carry a fusion in positive patients. We have reinforced the point that the fusions may be subclonal in the main text:

It is notable that patient samples carrying ostensibly the same SLC25A40-ABCB1 fusion event differed in the overall level of ABCB1 expression, suggesting that fewer tumour cells carried the fusion event in ascites with lower ABCB1 expression. (Line 72-74). The presence of subclonal fusion events suggests that in fusion-negative metastases novel mechanisms of ABCB1 deregulation or other mechanisms conferring resistance are extant. (Line 93-95)

Points 3 and 4. Cell line models.

The authors have not re-expressed fusion cDNA in MDR1-null cells to confirm the functional effect. However, the cell lines depicted in 4D suggest that the fusions can be functional. However, it is not clear which cases these cells lines derive from (and thus precisely which fusion is expressed in AOCs18.5 cells). This is not clear from the nomenclature nor supplementary information. However, these cells should act as an important model for acquired drug resistance.

The first line of Figure 4 legend and the Supplementary information both state that the fusion in AOCs18.5 is an *SLC25A40-ABCB1* fusion. AOCs18.5 was derived from Patient 15 and AOCs21.2 was derived from Patient 24, this has been added to the Supplementary Information.

The use of a novel high grade serous ovarian cancer patient derived cell line with the fusion surpasses expression of a cDNA of *ABCB1*, which has been done many times before. For example, the second sentence of the abstract of Robey et al *Nature Reviews Cancer* 2018 article on multidrug resistance sums up the situation with respect to involvement of *ABCB1* in chemoresistance (<https://doi.org/10.1038/s41568-018-0005-8>). The data is entirely consistent with the extensive known biology of *ABCB1/MDR1*.

In the manuscript we have added a sentence noting that:

Further functional characterization of the impact of other promoter fusion partners on chemosensitivity awaits the availability of appropriate cell lines with these fusion events. (Line 150-51)

Original minor points

Both of these have been addressed.

REVIEWERS' COMMENTS:

Reviewer #3 (Remarks to the Author):

The authors have addressed the comments clearly.